# Cuproof: Range Proof with Constant Size

**DOI:** 10.3390/e24030334

**Published:** 2022-02-25

**Authors:** Cong Deng, Lin You, Xianghong Tang, Gengran Hu, Shuhong Gao

**Affiliations:** 1School of Communication Engineering, Hangzhou Dianzi University, Hangzhou 310018, China; mrdengcong@gmail.com (C.D.); tangxh@hdu.edu.cn (X.T.); 2School of Cyberspace Security, Hangzhou Dianzi University, Hangzhou 310018, China; grhu@hdu.edu.cn; 3Department of Mathematical Sciences, Clemson University, Clemson, SC 29634, USA; sgao@clemson.edu

**Keywords:** zero-knowledge proof, range proof, inner-product, Bulletproofs, blockchain

## Abstract

Zero-Knowledge Proof is widely used in blockchains. For example, zk-SNARK is used in Zcash as its core technology to identifying transactions without the exposure of the actual transaction values. Up to now, various range proofs have been proposed, and their efficiency and range-flexibility have also been improved. Bootle et al. used the inner product method and recursion to construct an efficient Zero-Knowledge Proof in 2016. Later, Benediky Bünz et al. proposed an efficient range proof scheme called Bulletproofs, which can convince the verifier that a secret number lies in $\left[0,\;2^{\kappa }-1\right]$ with κ being a positive integer. By combining the inner-product and Lagrange’s four-square theorem, we propose a range proof scheme called Cuproof. Our Cuproof can make a range proof to show that a secret number *v* lies in an interval [a,b] with no exposure of the real value *v* or other extra information leakage about *v*. It is a good and practical method to protect privacy and information security. In Bulletproofs, the communication cost is 6+2logκ, while in our Cuproof, all the communication cost, the proving time and the verification time are of constant sizes.

## 1. Introduction

The blockchain technology is the most well-known decentralized and tamper-proof information technology, and it can be applied to construct many different digital service systems or application platforms, such as digital currencies, supply systems and so on. Wu et al. [1] elaborated the intellectual cores of the blockchain-Internet of Things (BIoT). Fedorov et al. [2] stated how to apply blockchain technology to 5G. Cryptocurrencies were the first to bring the concept of blockchain into the world. The blockchain-based cryptocurrencies enable peer-to-peer transactions and make sure that the transactions are valid. In the Bitcoin [3] system, all the transactions are recorded in a public ledger, and everyone can check whether the transactions in the ledger are valid. The hash function used in the blockchains ensures that the transaction data cannot be tampered with. However, every coin has two sides. Despite its advantage, the transparency in Bitcoin also has a disadvantage. In a transaction of Bitcoin, the transaction data, the addresses of the senders and the receivers are almost transparent, and it means that Bitcoin cannot achieve anonymity and cannot provide the same level of privacy as paper cash.

In order to offset the disadvantages that exist in Bitcoin, people have start to think about using zero-knowledge proof to protect the privacy of blockchain users, because a zero-knowledge proof is a cryptographic protocol that has strong privacy protection function. In [4], Sun et al. showed how zero-knowledge proof technology is applied to the blockchain. There are lots of blockchain-based cryptocurrencies using range proofs [5,6] or zk-SNARKs [7,8,9,10] such as Zcash [11]. The transactions between the shielded addresses are what makes Zcash special. In these transactions, although the traders’ addresses and the amount of the transactions are all covert, the validity of these transactions can still be checked because zk-SNARKs have been applied. According to the property of protecting anonymity, more and more cryptocurrencies apply range proof as a tool to avoid the disclosure of users’ information.

In 2018, Bünz et al. proposed a type of range proof that is called Bulletproofs [5]. The efficiency of Bulletproofs is particularly well suited for the blockchains. However, its communication cost, which is 6+2logκ, grows with larger κ. In this paper, we combine the Lagrange’s four-square theorem with Bulletproofs [5] to construct a range proof for arbitrary interval [a,b]. In our scheme, the communication cost is 4 elements of G and 18 elements of Z. Our Cuproof is a good method to protect uers’ privacy and information security. For example, we can use the Cuproof scheme to declare that our age *v* lies in some interval. Because of the RSA assumption and discrete logarithm problems, it is hard for the verifier to get the secret *v* but still believe that *v* is in this interval.

### 1.1. Related Work

Nowadays, information security or privacy protection has become more and more important for each of us. A number of works on information security or privacy protect have been published. For example, Dong et al. [12] elaborated how overconfidence affects information security investment and information security performance. Range proof technology, a kind of zero-knowledge proof protocol, is a good method for protecting information security or privacy. There have been lots of research works on range proof since the first relevant algorithm of range proof was proposed. Brickel et al. [6] first stated the correlative algorithm of range proof in 1987. Its purpose was to send reliable values to other participants, which can allow a user with a discrete logarithm to disclose one bit of information to another user so that any other user can verify the equations as they receive each bit. In 1998, Chan et al. [13] showed how to use the algorithm given in [6] to verify the non-negative transaction amount and they also enhanced the algorithm in [6]. Their improved proof method was called CTF proof. In 2000, Boudot [14] used the square numbers to build an effective range proof which was based on CTF.

By using the Lagrange’s four-square theorem [15], that is, any non-negative integer can be represented as the sum of squares of four integers, Lipmaa [16] proposed a proof of any range for the first time. In 2005, Groth [17] pointed out that if *y* is a non-negative integer, then 4y+1 could be represented as the sum of the squares of three integers. Using Boneh-Boyen signature [18], Teranishi et al. [19] proposed many anonymous authentication methods in 2006. In 2008, Camenisch et al. [20] used signature method that relies on the security of the q-Strong Diffie-Hellman assumptions to construct a range proof. In 2014, Belenkiy [21] designed a scheme to extend the u-proof cryptographic specification [22] by making use of the membership proof of a set. This scheme can be used twice to compare the size of one committed value with some other committed value, and therefore it can be used to construct a range proof.

Bootle et al. [23] made a step forward on the efficiency of space in Zero-Knowledge Proof based on discrete logarithms. They combined the inner product method and recursion to enhance the efficiency of Zero-Knowledge Proof. Based on this work, Bünz et al. [5] improved the inner product method for zero certificate range proof and proposed a more efficient Zero-Knowledge Proof scheme called Bulletproofs.

### 1.2. Contributions

Our scheme, called Cuproof for conveniency, is established on the techniques of Bulletproofs and Lagrange’s three-square theorem given in [17]. Our protocol can be used to construct a range proof for arbitrary range. The argument of our scheme has low computation complexity. The main difference between Bulletproofs and ours is that Bulletproofs’s communication cost [5] is logarithmic in κ, where κ is the exponent in the proving range $\left[0,2^{\kappa }-1\right]$, while the cost in our scheme is constant. The key is that we combine the following Theorem 2 with Bulletproofs. Our Cuproof satisfies the three security properties required for a secure Zero-Knowledge Proof: completeness, soundness, and zero-knowledge.

### 1.3. Structure of the Paper

In Section 2, some mathematical symbols, definitions, and theorems are given. The framework and construction of our range proof protocol are stated in Section 3. In Section 3.1, we show how to construct a proof that convinces the verifier that the prover knows the secret number *v*. In Section 3.2, we describe our range proof protocol Cuproof in detail. The performance comparisons among Bulletproofs, some other range proof protocols and Cuproof are shown in Section 4. Finally, the proof of Theorem 3 about our Cuproof will be given in Appendix A.

## 2. Preliminaries

Before we state our protocol, we first state some of the underlying tools. In this paper, A is a PPT adversary, which is a probabilistic interactive Turing Machine that runs in polynomial time in the security parameter λ.

### 2.1. Notation

Let [N] denote the set {1,…,N−1}. Let pandq denote two prime numbers. Let G denote the multiplicative group of integers modulo *n*, where *n* is the product of pandq, i.e., G is a RSA group. Let Z denote the set of all integers. Let $Z_{n}$ denote the ring of integers modulo *n*. Let $G^{j}$ and $Z_{n}^{j}$ be vector spaces of dimension *j* over G and $Z_{n}$, respectively. Let $Z_{n}^{*}$ denote $Z_{n}\setminus \left\{0\right\}$. Group elements which represent commitments are capitalized. For example, $C=g^{a}h^{\alpha }$ is a Pedersen commitment to *a* for g,h∈G. $x\overset{\$}{\leftarrow }Z_{n}^{*}$ means the uniform sampling of an element from $Z_{n}^{*}$. In this paper, $a\in F^{j}$ is a vector with elements $a_{1},\ldots ,a_{j}\in F$. For an element $c\in Z_{n}$ and a vector $a\in Z_{n}^{j}$, we denote by $b=c\cdot a\in Z_{n}^{j}$ the vector with $b_{i}=c\cdot a_{i}$. For the two vectors $a,b\in F^{j}$, let $\langle a,b\rangle =\sum_{i=1}^{j}a_{i}\cdot b_{i}$ denote the inner product and $a\circ b=\left(a_{1}\cdot b_{1},\ldots ,a_{j}\cdot b_{j}\right)\in F^{j}$ be the Hadamard product, respectively. We define vector polynomials $P\left(x\right)=\sum_{i=0}^{d}p_{i}\cdot x^{i}\in Z^{j}\left[x\right]$ where each coefficient $p_{i}$ is a vector in $Z^{j}$. The inner product between two vector polynomials l(x)andr(x) is defined as
(1)$$\langle l\left(x\right),r\left(x\right)\rangle =\sum_{i=0}^{d}\sum_{j=0}^{i}\langle l_{i},r_{j}\rangle \cdot x^{i+j}\in Z\left[x\right]$$
Let a∥b denote the concatenation of two vectors: if $a\in Z_{n}^{j}$ and $b\in Z_{n}^{m}$ then $a∥b\in Z_{n}^{j+m}$. For 0⩽ℓ⩽s, we use Python notation to denote slices of vectors:$$a_{\left[:\ell \right]}=a_{\left[0:\ell \right]}=\left(a_{1},\ldots ,a_{\ell }\right)\in F^{\ell },$$
$$a_{\left[\ell :\right]}=a_{\left[\ell :s\right]}=\left(a_{\ell +1},\ldots ,a_{s}\right)\in F^{s-\ell }.$$
Let t(x)=〈l(x),r(x)〉, then the inner product is defined such that t(x)=〈l(x),r(x)〉 holds for all $x\in Z_{n}$. For vectors $g=\left(g_{1},\ldots ,g_{j}\right)\in G^{j}$ and $a\in Z_{n}^{j}$, we write $C=g^{a}=\prod_{i=1}^{j}g_{i}^{a_{i}}\in G$. We set $\vec{u}=\left(1,2,3,\ldots ,u\right)\in Z^{u}$ for u≥1.

### 2.2. Assumptions

Groups of Unknown Order: In order to achieve the soundness of our range proof, we use the RSA group G where the order of the group is unknown. The RSA group is generated by a trusted setup.

RSA Group: In the multiplicative group G of the integers modulo *n* where *n* is the product of the large primes *p* and *q*. The hardness of computing the order of the group G is the same as the hardness of factoring *n*.

**Assumption** **1** (Discrete Log Relation Assumption)**.**
*For all PPT adversaries A and j≥2, there exists a negligible function μ(λ) such that:*

$$\begin{array}{c} P\left[\begin{array}{c} G=\mathrm{Setup}\left(1^{\lambda }\right),\\ g_{1},\ldots ,g_{j}\overset{\$}{\leftarrow }G;\\ a_{1},\ldots ,a_{j}\in Z_{2^{\lambda }n}\leftarrow A\left(g_{1},\ldots ,g_{j}\right) \end{array}:\begin{array}{c} \exists a_{i}\neq 0,\\ \prod_{i=1}^{j}g_{i}^{a_{i}}=1 \end{array}\right]\leq \mu \left(\lambda \right). \end{array}$$



As Bünz et al. [5] stated, $\prod_{i=1}^{j}g_{i}^{a_{i}}=1$ is a non trivial discrete log relation among $g_{1},\ldots ,g_{j}$. The discrete log relation assumption makes sure that an adversary cannot find a non-trivial relation between randomly selected group elements. This assumption is equivalent to the discrete-log assumption when j≥1.

**Assumption** **2** (Order Assumption)**.**
*For any efficient adversary A there exists a negligible function μ(λ) such that:*

Pg1≠1∧g1a1=1:G←$Setup(λ),(g1,a1)←$A(G),wherea1≠0∈Z2λn,andg1∈G≤μ(λ).



**Lemma** **1.**
*A PPT adversary A breaking Order Assumption can also break Discrete Log Relation Assumption easily.*


**Proof.** We show that if an adversary $A_{Ord}$ breaks the Order Assumption, then we can construct $A_{DL}$ which breaks the Discrete Log Relation Assumption with overwhelming probability. In order to get a vector $\left(g_{1},g_{2},\ldots ,g_{j}\right)\in G^{j}$ and a vector $\left(a_{1},a_{2},\ldots ,a_{j}\right)\in Z_{2^{\lambda n}}^{j}$ such that $g_{1}^{a_{1}}\cdot g_{2}^{a_{2}}\cdots g_{j}^{a_{j}}=1$ where $g_{i}\neq 1,a_{i}\neq 0$ and i∈{1,2,…,j}, we run $A_{Ord}$ for *n* times and it will output $g_{j}\in G$ and $a_{j}\in Z$ such that $g_{j}^{a_{j}}=1$ for j=1,…,n. It follows that $\prod_{j=1}^{n}g_{j}^{a_{j}}=1$. □

### 2.3. Commitments

**Definition** **1** (Commitments)**.**
*A non-interactive commitment scheme consists of a pair of probabilistic polynomial time algorithms (Setup,Com). The setup algorithm $pp\leftarrow \mathrm{Setup}(1^{\lambda })$ generates the public parameters pp with the security parameter λ. The commitment algorithm ${\mathrm{Com}}_{pp}$ defines a function $M_{pp}\times R_{pp}\to C_{pp}$ for a message space $M_{pp},$ a randomness space $R_{pp}$, and a commitment space $C_{pp}$ determined by pp. For a message $x\in M_{pp},$ the algorithm draws $r\overset{\$}{\leftarrow }R_{pp}$ uniformly at random, and computes commitment $\mathrm{com}={\mathrm{Com}}_{pp}\left(x,r\right).$*


**Definition** **2** (Pedersen Commitment)**.**
*Let $M_{pp}=Z_{n},R_{pp}=Z_{2^{\lambda }n}\;\mathrm{and}\;C_{pp}=\left(G,*\right)$ be a multiplicative group, the commitment is generated as follows:*

$$\begin{array}{c} \mathrm{Setup}:g,h\overset{\$}{\leftarrow }G,\\ \mathrm{Com}\left(x;r\right)=\left(g^{x}h^{r}\right). \end{array}$$



**Definition** **3** (Pedersen Vector Commitment)**.**
*Let $M_{pp}=Z_{n}^{j},R_{pp}=Z_{2^{\lambda }n}\;\mathrm{and}\;C_{pp}=\left(G,*\right)$ being a multiplicative group, the commitment is generated as follows:*

$$\begin{array}{c} \mathrm{Setup}:g=\left(g_{1},\ldots ,g_{j}\right),h\overset{\$}{\leftarrow }G,\\ \mathrm{Com}\left(x=\left(x_{1},\ldots ,x_{j}\right);r\right)=h^{r}g^{x}=h^{r}\prod_{i}g_{i}^{x_{i}}\in G. \end{array}$$



### 2.4. Zero-Knowledge Arguments of Knowledge

A Zero-Knowledge Argument consists of three interactive algorithms (Setup, P, V) which run in probabilistic polynomial time. Setup is the common reference string generator, P is the prover, and V is the verifier. The algorithm Setup produces a common reference string σ on inputting $1^{\lambda }$. The transcript produced by P and V is denoted by tr←<P(s),V(t)> when they interact on the inputs *s* and *t*. We write <P(s),V(t)>=b where b=0 if the verifier rejects, b=1 if the verifier accepts.

Let R be a polynomial-time-decidable ternary relation. Given a parameter σ, the *w* is a witness for a statement *u* only if (σ,u,w)∈R. We define the CRS-dependent language
$$L_{\sigma }=\left\{u|\exists w:\left(\sigma ,u,w\right)\in R\right\}$$
as the set of all the statements which have a witness *w* in the relation R.

**Definition** **4** (Argument of Knowledge)**.**
*(Setup,P,V) is called an argument of knowledge for relation R if it satisfies both the Perfect Completeness and the Computational Soundness.*

*Perfect Completeness:*

$$P\left[\begin{array}{c} \left(\sigma \right)\leftarrow \mathrm{Setup}\left(1^{\lambda }\right);\\ \left(u,w)\leftarrow A(\sigma \right) \end{array}|\begin{array}{c} (\sigma ,u,w)\notin Ror\\ \langle P(\sigma ,u,w),V(\sigma ,u)\rangle =1 \end{array}\right]=1.$$


*Computational Soundness:*

$$P\left[\begin{array}{c} \left(\sigma \right)\leftarrow \mathrm{Setup}\left(1^{\lambda }\right);\\ u\leftarrow A(\sigma ) \end{array}|\begin{array}{c} u\notin L_{\sigma }and\\ \langle A,V(\sigma ,u)\rangle =1 \end{array}\right]\approx 0.$$



**Definition** **5** (Perfect Special Honest-Verifier Zero-Knowledge)**.**
*A public coin argument of knowledge (Setup,P,V), as defined in [5], is a perfect special honest verifier zero knowledge (SHVZK) argument of knowledge for R if there exists a probabilistic polynomial time simulator S such that for every pair of interactive adversaries $A_{1}\;\mathrm{and}\;A_{2}$, we have*

$$\begin{array}{c} P\left[\begin{array}{cc} \left(\sigma ,u,w\right)\in R\;\mathrm{and}\;A_{1}\left(tr\right)=1 & |\begin{array}{c} \sigma \leftarrow \mathrm{Setup}(1^{\lambda })\\ \left(u,w,\rho \right)\leftarrow A_{2}\left(\sigma \right)\\ tr\leftarrow \langle P(\sigma ,u,w)\\ V(\sigma ,u;\rho )\rangle \end{array} \end{array}\right]\\ =P\left[\begin{array}{cc} \left(\sigma ,u,w\right)\in R\;\mathrm{and}\;A_{1}\left(tr\right)=1 & |\begin{array}{c} \sigma \leftarrow \mathrm{Setup}(1^{\lambda })\\ \left(u,w,\rho \right)\leftarrow A_{2}\left(\sigma \right)\\ tr\leftarrow S(u,\rho ) \end{array} \end{array}\right] \end{array}$$

*where ρ is the public coin randomness used by the verifier. The "transcript" can be simulated by S without knowing w.*


**Definition** **6** (Zero-Knowledge Range Proof)**.**
*Given a commitment scheme (Setup,Com) over a message space $M_{pp}$ which is a set with a total ordering, a Zero-Knowledge range proof is a SHVZK argument of knowledge for the relation $R_{\mathrm{Range}}:$*

$$\begin{array}{c} \left(pp,\left(\mathrm{com},l,r\right),\left(x,\rho \right)\right)\in R_{\mathrm{Range}}\\ ⇕\\ \mathrm{com}=\mathrm{Com}(x;\rho )\;\wedge \;(l\leq x<r). \end{array}$$



**Theorem** **1** (Lagrange’s four-square theorem)**.**
*Any non-negative integer can be represented as the sum of the squares of four integers.*


The proof for Theorem 1 is given in [15] and an algorithm for finding four such squares was provided in [16].

**Theorem** **2** (Lagrange’s three-square theorem)**.**
*If x is a positive integer, then 4x+1 can be written as the sum of three integer squares.*


The proof for Theorem 2 is given in [17], and ref. [15] offered an efficient and simple algorithm for finding three such squares. Theorem 2 also means writing 4x+1 as the sum of three squares implies that *x* is non-negative.

## 3. Efficient Range Proof Protocol

In this section, we will present our range proof protocol.

### 3.1. Four Integer Zero-Knowledge Proof

We now describe how to use the inner-product argument to construct a proof. The prover convinces the verifier that a commitment *V* contains a number *v* in a given range without revealing *v*.

In our proof, a Pedersen commitment *V* is an element in the group G that is used to perform the inner product argument and λ is the security parameter.

We let $v\in Z_{n}$, and an element V∈G be a Pedersen commitment to *v* which uses a random number *r*. The proof system proves the following relation:(2)$$\;\{\left(g,h,V\in G;\;v\in Z_{n},\;r\in Z_{2^{\lambda }n}\right):\;V=h^{r}g^{v}\}$$

Choose $a=\left(a_{1},a_{2},a_{3},a_{4}\right)\in Z_{n}^{4}$ such that
(3)$$v=a_{1}^{2}+a_{2}^{2}+a_{3}^{2}+a_{4}^{2},\;i.e.\;\langle a,a\rangle =v$$
Let $y\in Z_{2^{\lambda }n}^{*}$ and $y=\vec{4}\cdot y\in Z^{4}$. The prover P uses an element in G to generate a commitment to the vector a. To convince V that *v* be a positive number, the prover must prove that he knows an opening $a\in Z_{n}^{4}$ satisfying 〈a,a〉=v. To construct this zero knowledge proof, V should randomly choose $z\in Z_{2^{\lambda }n}$, and then the prover proves that
(4)$$\langle a,a\rangle z^{2}+\langle a-a,y\rangle z=vz^{2}$$
This equality can be re-written as:(5)$$\langle a\cdot z-y,\;a\cdot z+y\rangle =vz^{2}-\delta \left(y\right)$$
The verifier can easily calculate that δ(y)=〈y,y〉∈Z. Hence, the problem of proving that Equation (Equation 3) holds is reduced to proving a single inner-product identity.

If the prover sends to the verifier the two vectors in the inner product in Equation (Equation 5), then the verifier could check Equation (Equation 5) itself by using the commitment *V* to *v* and be convinced that Equation (Equation 3) holds. However, these two vectors reveal the information of a and so the prover cannot send them to the verifier. To solve this problem, we use two additional blinding terms $s_{L},s_{R}\in Z_{2^{\lambda }n}^{4}$.

To prove the statement Equation (Equation 2), P and V should obey the following protocol:$$\begin{array}{cc} P & \;\mathrm{inputs}\;v,r\;\mathrm{and}\;\mathrm{computes}: \end{array}$$(6)$$\begin{array}{cc} \; & \;a=\left[a_{1},a_{2},a_{3},a_{4}\right]\in Z_{n}^{4}\;s.t.\langle a,a\rangle =v \end{array}$$(7)$$\begin{array}{cc} \; & \;\alpha \overset{\$}{\leftarrow }Z_{2^{\lambda }n} \end{array}$$(8)$$\begin{array}{cc} \; & \;A=h^{\alpha }g^{a}h^{a}\in G \end{array}$$(9)$$\begin{array}{cc} \; & \;s_{L},s_{R}\overset{\$}{\leftarrow }Z_{2^{\lambda }n}^{4} \end{array}$$(10)$$\begin{array}{cc} \; & \;\rho \overset{\$}{\leftarrow }Z_{2^{\lambda }n} \end{array}$$(11)$$\begin{array}{cc} \; & \;S=h^{\rho }g^{s_{L}}h^{s_{R}}\in G \end{array}$$(12)$$\begin{array}{cc} P & \;\to V:A,S \end{array}$$(13)$$\begin{array}{cc} V & \;:y^{\prime },z^{\prime }\overset{\$}{\leftarrow }Z_{2^{\lambda }n}^{*} \end{array}$$(14)$$\begin{array}{cc} V & \;\mathrm{computes}\;:y=g^{y^{\prime }},z=g^{z^{\prime }}\in G \end{array}$$(15)$$\begin{array}{cc} V & \;\to P:y,z \end{array}$$
Here, let us expand two linear vector polynomials l(x)andr(x) in $Z^{4}\left[x\right]$, and a quadratic polynomial t(x)∈Z[x] as follows:$$\begin{array}{cc} l\left(x\right)=az-y+s_{L}x & \;\;\;\;\in Z^{4}\left[x\right]\\ r\left(x\right)=az+y+s_{R}x & \;\;\;\;\in Z^{4}\left[x\right]\\ t\left(x\right)=\langle l\left(x\right),r\left(x\right)\rangle =t_{0}+t_{1}\cdot x+t_{2}\cdot x^{2} & \in Z[x] \end{array}$$
The constant terms of l(x)andr(x) are the inner product vectors in Equation (Equation 5). The blinding vectors $s_{R}\;\mathrm{and}\;s_{L}$ make sure that the prover can publish l(x)andr(x) for random *x* and does not need to reveal any information of a. The constant term $t_{0}$ of t(x) is the result of the inner product in Equation (Equation 5). The prover needs to convince the verifier that the following equation hold:$$t_{0}=vz^{2}-\delta \left(y\right)$$
$$\begin{array}{cc} P & \;\mathrm{computes}: \end{array}$$
(16)$$\begin{array}{cc} \; & \;\tau_{1},\tau_{2}\overset{\$}{\leftarrow }Z_{2^{\lambda }n} \end{array}$$
(17)$$\begin{array}{cc} \; & \;T_{i}=g^{t_{i}}h^{\tau_{i}}\in G,\;\;i\in \left\{1,2\right\} \end{array}$$
(18)$$\begin{array}{cc} P & \;\to V:T_{1},T_{2} \end{array}$$
(19)$$\begin{array}{cc} V & \;:x^{\prime }\overset{\$}{\leftarrow }Z_{2^{\lambda }n}^{*} \end{array}$$
(20)$$\begin{array}{cc} V & \;\mathrm{computes}:x=g^{x^{\prime }}\in G \end{array}$$
(21)$$\begin{array}{cc} V & \;\to P:x\\ P & \;\mathrm{computes}: \end{array}$$
(22)$$\begin{array}{cc} \; & \;l=l\left(x\right)=az-y+s_{L}x\;\in Z^{4} \end{array}$$
(23)$$\begin{array}{cc} \; & \;r=r\left(x\right)=az+y+s_{R}x\;\in Z^{4} \end{array}$$
(24)$$\begin{array}{cc} \; & \;\hat{t}=\langle l,r\rangle \in Z \end{array}$$
(25)$$\begin{array}{cc} \; & \;\tau_{x}=\tau_{2}\cdot x^{2}+\tau_{1}\cdot x+z^{2}r\;\in Z \end{array}$$
(26)$$\begin{array}{cc} \; & \;\mu =\alpha z+\rho x\;\in Z \end{array}$$
(27)$$\begin{array}{cc} P & \;\to V:\tau_{x},\mu ,\hat{t},l,r \end{array}$$
$$\begin{array}{cc} V & \;\mathrm{checks}\;\mathrm{these}\;\mathrm{equations}\;\mathrm{and}\;\mathrm{computes}: \end{array}$$
(28)$$\begin{array}{cc} \; & \;P=A^{z}\cdot S^{x}\cdot g^{-y}\cdot h^{y}\;\in G \end{array}$$
(29)$$\begin{array}{cc} \; & \;P\overset{?}{=}h^{\mu }\cdot g^{l}\cdot h^{r}\;\in G \end{array}$$
(30)$$\begin{array}{cc} \; & \;g^{\hat{t}}h^{\tau_{x}}\overset{?}{=}V^{z^{2}}g^{-\delta (y)}\cdot T_{1}^{x}\cdot T_{2}^{x^{2}}\in G \end{array}$$
(31)$$\begin{array}{cc} \; & \;\hat{t}\overset{?}{=}\langle l,r\rangle \in Z \end{array}$$

**Corollary** **1** (Four-Integer Zero-Knowledge Proof)**.**
*The Four-Integer Zero-Knowledge Proof presented in Section 3.1 has perfect completeness, perfect special honest verifier zero-knowledge, and computational soundness.*


**Proof.** The Four-Integer Zero-Knowledge Proof is a special case of the aggregated logarithmic proof from the following Section 3.2 with m=1, hence, it is a direct corollary of Theorem 3. □

### 3.2. Aggregating Logarithmic Proofs

Bünz et al. [5] stated a type of proof for *m* values, which is more efficient than conducting *m* individual range proofs. Based on Bulletproofs, we can also perform a proof for *m* values as [5] does. In this section, we show that this can be done with some modification to the protocol of zero-knowledge proof in Section 3.1. The relation that we will prove is as follows:(32)$$\begin{array}{c} \{\left(g,h\in G,V\in G^{m};v\in Z_{n}^{m},\;r\in Z_{2^{\lambda }n}^{m}\right):\\ V_{j}=h^{r_{j}}g^{v_{j}}\;\mathrm{for}\mathrm{all}\;j\in \left[m\right]\}. \end{array}$$

The prover does similar work as the prover does for a simple zero-knowledge proof in Section 3.1 except for the following modifications. First, we set $y\in Z_{2^{\lambda }n}^{*},y=y\cdot \vec{4m}\in Z^{4m}\;\mathrm{and}\;\left|\vec{4m}\right|=4m$. As in Equation (Equation 6), the prover needs to find $a\in Z_{n}^{4m}$ so that
$$\langle a_{\left[4(j-1):4j\right]},a_{\left[4(j-1):4j\right]}\rangle =v_{j}\;\mathrm{for}\mathrm{all}\;j\in \left[m\right].$$
We accordingly modify l(x) and r(x) as follows:(33)$$l\left(x\right)=\sum_{j=1}^{m}z\cdot j\left(0^{4(j-1)}∥a_{\left[4(j-1):4j\right]}∥0^{4(m-j)}\right)-y+s_{L}\cdot x$$
(34)$$r\left(x\right)=\sum_{j=1}^{m}z\cdot j\left(0^{4(j-1)}∥a_{\left[4(j-1):4j\right]}∥0^{4(m-j)}\right)+y+s_{R}\cdot x$$
To compute $\tau_{x}$, we adjust the randomness $r_{j}$ of each commitment $V_{j}$ such that $\tau_{x}=\tau_{1}\cdot x+\tau_{2}\cdot x^{2}+z^{2}\sum_{j=1}^{m}j^{2}\cdot r_{j}$. That is, the verification checking Equation (Equation 30) needs to be adjusted to include all the $V_{j}$ commitments as follows
(35)$$g^{\hat{t}}h^{\tau_{x}}=V^{\left(z^{2}\cdot \vec{m}\circ \vec{m}\right)}g^{-\delta (y)}T_{1}^{x}T_{2}^{x^{2}}$$
Finally, we change the definition of *A* as follows:(36)$$A=h^{\alpha }\prod_{j=1}^{m}g_{\left[4(j-1):4j\right]}^{j\cdot a_{\left[4(j-1):4j\right]}}\cdot \prod_{j=1}^{m}h_{\left[4(j-1):4j\right]}^{j\cdot a_{\left[4(j-1):4j\right]}}$$

**Theorem** **3** (Aggregate Logarithmic Proof)**.**
*The Aggregate Logarithmic Proof presented in Section 3.2 has perfect completeness, perfect honest verifier zero-knowledge, and computational soundness.*


The proof for Theorem 3 is presented in Appendix A. This protocol can also be transformed into a NIZK protocol by using the Fiat-Shamir heuristic.

### 3.3. Our Protocol: Cuproof

In this section, we will demonstrate how to prove that a secret number is within an arbitrary interval. The goal of our range proof protocol is to convince the verifier that the secret number *v* is in [a,b]. Based on Theorem 2, We can find $a,b\in Z_{n}$ and $d=\left(d_{1},\ldots ,d_{6}\right)\in Z_{n}^{6}$ such that the following conditions hold:(37)$$\left\{\begin{array}{c} d_{1}^{2}+d_{2}^{2}+d_{3}^{2}=4v-4a+1=v_{1}\in Z,\\ d_{4}^{2}+d_{5}^{2}+d_{6}^{2}=4b-4v+1=v_{2}\in Z. \end{array}\right.$$
The whole protocol is similar to the special case of the aggregating logarithmic proofs from Section 3.2 for m=2 and $a\in Z_{n}^{6}$. In this protocol, we set $\delta \left(y\right)\in Z,\;y\in Z^{6}$. We will prove the following relations:(38)$$\begin{array}{c} \{\left(g,h\in G,V=\left(V_{1},V_{2}\right)\in G^{2}\right):\\ V_{j}=h^{r_{j}}g^{v_{j}}\;\forall \;j\in \left\{1,2\right\},V=g^{v}h^{r}\wedge v\in \left[a,b\right]\} \end{array}$$
The protocol is as follows:$$\begin{array}{cc} P & \;\mathrm{inputs}\;v,r\;\mathrm{and}\;\mathrm{computes}: \end{array}$$(39)$$\begin{array}{cc} \; & \;v_{1}=4v-4a+1,v_{2}=4b-4v+1\;\in Z, \end{array}$$(40)$$\begin{array}{cc} \; & \;\mathrm{Finds}\;d=\left(d_{1},\ldots ,d_{6}\right)\;\mathrm{satisfying}\;\left(37\right) \end{array}$$(41)$$\begin{array}{cc} \; & \;\alpha \overset{\$}{\leftarrow }Z_{2^{\lambda }n} \end{array}$$(42)$$\begin{array}{cc} \; & \;A=h^{\alpha }\prod_{j=1}^{2}g_{\left[3(j-1):3j\right]}^{j\cdot d_{\left[3(j-1):3j\right]}}\cdot \prod_{j=1}^{2}h_{\left[3(j-1):3j\right]}^{j\cdot d_{\left[3(j-1):3j\right]}}\in G \end{array}$$(43)$$\begin{array}{cc} \; & \;s_{L},s_{R}\overset{\$}{\leftarrow }Z_{2^{\lambda }n}^{6} \end{array}$$(44)$$\begin{array}{cc} \; & \;\rho \overset{\$}{\leftarrow }Z_{2^{\lambda }n} \end{array}$$(45)$$\begin{array}{cc} \; & \;S=h^{\rho }g^{s_{L}}h^{s_{R}}\in G \end{array}$$(46)$$\begin{array}{cc} P & \;\to V:A,S \end{array}$$(47)$$\begin{array}{cc} V & \;:y^{\prime },z^{\prime }\overset{\$}{\leftarrow }Z_{2^{\lambda }n}^{*} \end{array}$$(48)$$\begin{array}{cc} V & \;\mathrm{computes}\;:y=g^{y^{\prime }},z=g^{z^{\prime }}\in G \end{array}$$(49)$$\begin{array}{cc} V & \;\to P:y,z \end{array}$$
Here, as shown in Section 3.1, we have
$$t\left(x\right)=\langle l\left(x\right),r\left(x\right)\rangle =t_{0}+t_{1}\cdot x+t_{2}\cdot x^{2}\in Z\left[x\right].$$
$$\begin{array}{cc} P & \;\mathrm{computes}: \end{array}$$
(50)$$\begin{array}{cc} \; & \;\tau_{1},\tau_{2}\overset{\$}{\leftarrow }Z_{2^{\lambda }n} \end{array}$$
(51)$$\begin{array}{cc} \; & \;T_{i}=g^{t_{i}}h^{\tau_{i}}\in G,i\in \left\{1,2\right\}\\ \; & \;(t_{1},t_{2}\;\mathrm{can}\;\mathrm{be}\;\mathrm{computed}\;\mathrm{without}\;\mathrm{knowing}\;x) \end{array}$$
(52)$$\begin{array}{cc} P & \;\to V:T_{1},T_{2} \end{array}$$
(53)$$\begin{array}{cc} V & \;:x^{\prime }\overset{\$}{\leftarrow }Z_{2^{\lambda }n}^{*} \end{array}$$
(54)$$\begin{array}{cc} V & \;\mathrm{computes}:x=g^{x^{\prime }}\in G \end{array}$$
(55)$$\begin{array}{cc} V & \;\to P:x \end{array}$$
(56)$$\begin{array}{cc} P & \;\mathrm{computes}:\\ \; & \;l=z\cdot \sum_{j=1}^{2}j\cdot (0^{3(j-1)}∥d_{\left[3(j-1):3j\right]}∥0^{3(2-j)}) \end{array}$$
(57)$$\begin{array}{cc} \; & \;\;\;\;\;\;-y+s_{L}x\in Z^{6}.\\ \; & \;r=z\cdot \sum_{j=1}^{2}j\cdot (0^{3(j-1)}∥d_{\left[3(j-1):3j\right]}∥0^{3(2-j)}) \end{array}$$
(58)$$\begin{array}{cc} \; & \;\;\;\;\;\;+y+s_{R}x\in Z^{6}. \end{array}$$
(59)$$\begin{array}{cc} \; & \;\hat{t}=\langle l,r\rangle =t_{0}+t_{1}\cdot x+t_{2}\cdot x^{2}\in Z \end{array}$$
(60)$$\begin{array}{cc} \; & \;r_{1}=4r,r_{2}=-4r\;\in Z \end{array}$$
(61)$$\begin{array}{cc} \; & \;\tau_{x}=\tau_{2}x^{2}+\tau_{1}x+z^{2}\sum_{j=1}^{2}j^{2}\cdot r_{j}\in Z \end{array}$$
(62)$$\begin{array}{cc} \; & \;\mu =\alpha z+\rho x\in Z \end{array}$$
(63)$$\begin{array}{cc} P & \;\to V:\tau_{x},\mu ,\hat{t},l,r\\ V & \;\mathrm{computes}\;\mathrm{and}\;\mathrm{checks}\;\mathrm{these}\;\mathrm{equations}: \end{array}$$
(64)$$\begin{array}{cc} \; & \;V_{1}=V^{4}\cdot g^{-4a}\cdot g=g^{4v-4a+1}h^{4r}=g^{v_{1}}h^{r_{1}}\in G \end{array}$$
(65)$$\begin{array}{cc} \; & \;V_{2}=g^{4b}\cdot V^{-4}\cdot g=g^{4b-4v+1}h^{-4r}=g^{v_{2}}h^{r_{2}}\in G \end{array}$$
(66)$$\begin{array}{cc} \; & \;V=\left(V_{1},V_{2}\right)\;\in G^{2} \end{array}$$
(67)$$\begin{array}{cc} \; & \;P=A^{z}S^{x}g^{-y}h^{y}\;\in G \end{array}$$
(68)$$\begin{array}{cc} \; & \;P\overset{?}{=}h^{\mu }g^{l}h^{r}\;\in G \end{array}$$
(69)$$\begin{array}{cc} \; & \;g^{\hat{t}}h^{\tau_{x}}\overset{?}{=}V^{z^{2}\cdot \left(\vec{2}\circ \vec{2}\right)}g^{-\delta (y)}T_{1}^{x}T_{2}^{x^{2}}\;\in G \end{array}$$
(70)$$\begin{array}{cc} \; & \;\hat{t}\overset{?}{=}\langle l,r\rangle \;\in Z \end{array}$$

**Theorem** **4.**
*The protocol for range proof presented here above has perfect completeness, perfect special honest verifier zero-knowledge, and computational soundness.*


**Proof.** The protocol for range proof is a special case of the Aggregated Logarithmic Proof in Section 3.2 with m=2 and $a\in Z_{n}^{6}$. Hence, this theory is a direct corollary of our Theorem 3. □

In short, we call our given protocol for range proof *Cuproof*.

## 4. Performance

In order to evaluate the practical performance of our Cuproof, we provide a reference implementation in Python. We set that the sizes of the two primes *p* and *q* are 1024 bits. The prover uses the algorithms of [15,16] to generate the witnesses a and d, and compute the landr. A Pedersen hash function over an RSA group whose modulo n=p∗q is benchmarked. We performed our experiments on our computer with an Intel i5-7500 CPU@3.4 GHZ and we used a single thread. Table 1 shows the comparison of our Cuproof with Bulletproofs and the three range proofs put out by Boudot [14], Lipmaa [16] and Groth et al. [24], respectively. It states that the communication cost is const while Bulletproof’s communication cost is sublinear in *n*. Moreover, Cuproof is more efficient than the three range proof schemes proposed by Boudot [14], Lipmaa [16] and Groth et al. [24], respectively. Table 2 shows the proving time, verification time, and the gates of the range proofs under the different ranges (the final data is the average of the data we obtained by doing 10,000 experiments). Figure 1 shows the line charts of the proving time and the verification time of the Four-Integer Zero-Knowledge Proofs (no including the witness generation) for the secret of the different sizes, respectively. Figure 2 shows the line charts of the proving time and the verification time of the Range Proofs (no including witness generation), respectively. No matter how large the range is, the proving time is near 170 ms and the verification time is near 447 ms. Figure 3 shows the proof sizes in different intervals and it demonstrates that the proof size is near 5500 bytes. Table 3 shows the proof sizes, proving time and the verification time for the interval range proofs on the different sizes, respectively.

## 5. Conclusions

In this paper, we construct a kind of range proof scheme *Cuproof*, which can prove v∈[a,b] without revealing *v*’s actual value. In our protocol, by combining Theorem 2 into Bulletproofs, we reduce the communication cost to the constant sizes, make the computation complexity lower, and enhance the efficiency of our range proof. Compared to the works [14,16], our zero-knowledge proof *Cuproof* is more efficient. The Cuproof can be applied to cryptocurrencies such as Monero [25] does and it can also be used for personal privacy protection. For example, in a biometric-based identity authentication system, we can use our Cuproof to prove that the Euclidean distance between the two biometric vectors respectively extracted during the registration phase and during authentication phase is within a preset threshold to identify a user’s identity. Besides, we can also use Cuproof to prove that we are adults without exposing our true age. For instance, we can use Cuproof to prove that our age is lager than 18. However, a disadvantage of our range proof is that it still needs a trusted setup. Once the trusted setup is malicious, the secret number needs to be proved whether it has been leaked. In addition, because the security of Cuproof is based on the discrete logarithm problem, it is vulnerable to quantum attacks. Therefore, in our future work, we may use two groups to remove the trusted setting, one is a common group and the other is the verifier’s secret group, that is, Equation (Equation 68) is checked in the common group and Equation (Equation 69) is checked in the verifier’s secret group. In addition, in order to resist quantum attacks, we will consider to improve Cuproof based on an integer lattice. For example, we will use the elements in some integer lattice to replace the secret vectors of Cuproof. 

## Figures and Tables

**Figure 1 entropy-24-00334-f001:**
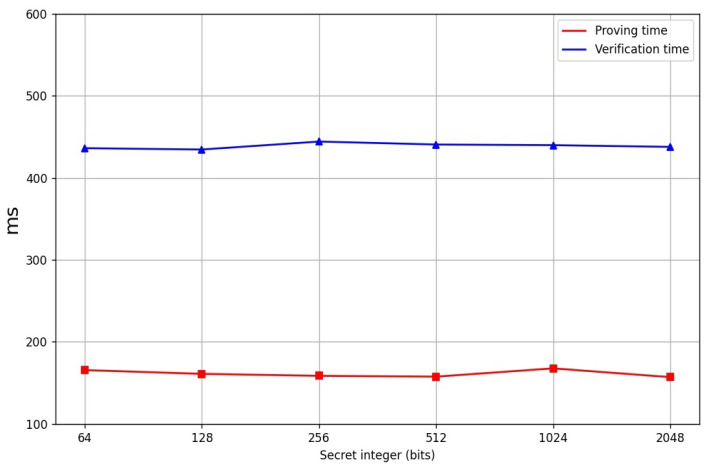
Four-integer zero-knowledge proof time.

**Figure 2 entropy-24-00334-f002:**
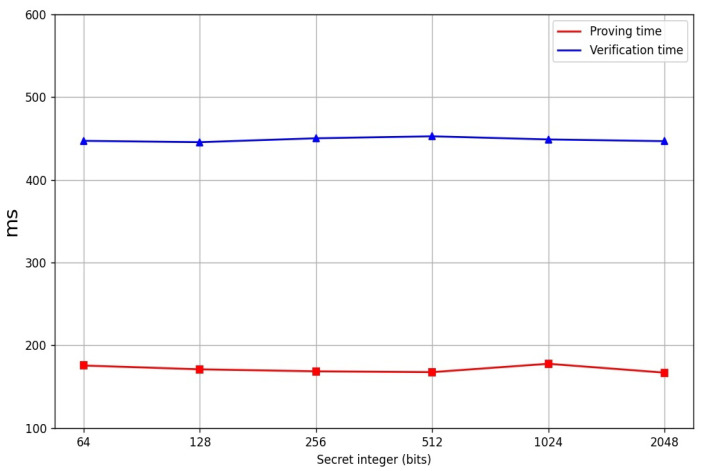
Range proof time.

**Figure 3 entropy-24-00334-f003:**
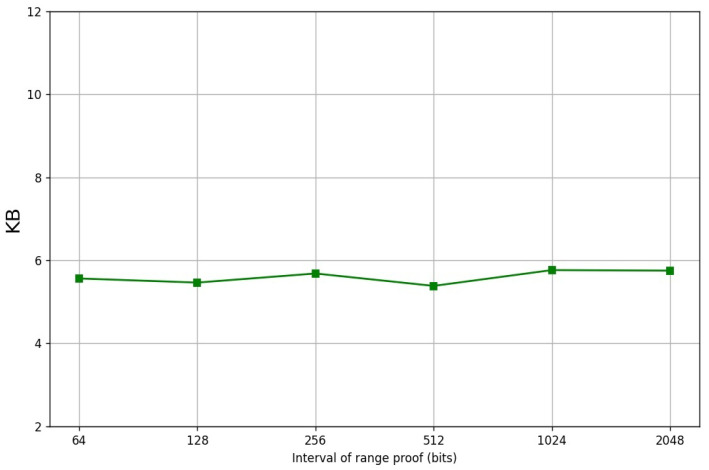
Sizes for range proofs.

**Table 1 entropy-24-00334-t001:** The comparison of Cuproof with Bulletproofs and the three range proofs respectively proposed by Boudot [14], Lipmaa [16] and Groth [24] for arithmetic circuit satisfiability with *d* the maximum size of the committed polynomials, *m* wires, SRS (the structured reference string) and *n* gates. The computational costs are measured in terms of the number of group elements and ring elements. mG means *m* group elements in the RSA group, ℓEx means *ℓ* group exponentiations. *ℓ* is the number of the elements that the known circuit inputs.

Scheme	Universal SRS	Circle SRS	Size	P’s Computation	V’s Computation
Bulletproofs [5]	$\frac{n}{2}G$	−	$2\log_{2}\left(n\right)+6G+5Z_{p}$	8nEx	4nEx
Boudot [14]	16G	−	6G+19Z	36Ex	38Ex
Lipmaa [16]	14G	−	12G+18Z	36Ex	36Ex
Groth et al. [24]	−	3n+mG	3G	4n+m−ℓEx	3P+ℓEx
This work	14G	−	7G+15Z	28Ex	38Ex

**Table 2 entropy-24-00334-t002:** Asymptotic efficiency comparison of zero-knowledge proofs for arithmetic circuits. Here *n* is the number of gates. A white rhombus for post-quantum security denotes that it is feasibly post-quantum secure. A black rhombus for untrusted setup denotes that the scheme is updatable. DL stands for discrete log.

Scheme	PQ?	Universal	Untrusted Setup	Assumption	Runtime	
					Prover	Verifier
Bulletproofs [5]	◊	⧫	⧫	DL	O(nlog(n))	O(nlog(n))
Boudot [14]	◊	⧫	◊	DL	O(nlog(2n))	O(nlog(n+2))
Lipmaa [16]	◊	⧫	◊	DL	O(nlog(2n+4))	O(nlog(2n))
This work	◊	⧫	◊	RSA	O(6log(n))	O(6log(n))

**Table 3 entropy-24-00334-t003:** Our Cuproof’s performances for the different sizes’ range proofs.

Range Size	Gates	Proof Size	Timing (ms)	
		(Bytes)	Prove	Verify
64bit	6	5561	175.4	446.2
128bit	6	5462	170.8	444.6
256bit	6	5681	168.4	452.3
512bit	6	5382	167.4	450.7
1024bit	6	5763	177.5	449.6
2048bit	6	5751	166.8	447.8

## Data Availability

Not applicable.

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
