# Peer review of "Cuproof: Range Proof with Constant Size"

_entropy, 2022, doi:10.3390/e24030334_

Round 1
Reviewer 1 Report
The article is an improvement of the protocol presented in the huge article Bunz, Bootley et al : "Bulletproofs: Short Proofs for Confidential Transactions and More". Blockchains offer a decentralized, immutable and verifiable ledger that can record transactions of digital assets. Privacy plays a central role in many application domains that utilize blockchain technology. However, blockchains are subject to potential privacy issues such as transaction linkability, compliance with data protection regulations, on-chain data privacy, and malicious smart contracts. Bulletproofs is a non-interactive zero-knowledge proof protocol with very short proofs and without a trusted setup; the proof size is logarithmic in the witness size. Bulletproofs are especially well suited for efficient range proofs on committed values.
The improvement of the article with a new Zero Knowledge crypto proof is due to the Lagrange theorem (if y is a non-negative integer, then 4y + 1 could be represented as the sum of the squares of three integers), combined with the Bulletproofs protocol.
Graphics and Tables are poor.
The approach of the improvement is original. We are waiting for the future work with a protocol without a trusted setup.
"In order to evaluate the practical performance of our Cuproof, we provide a reference implementation in Python" : not presented to the Reviewer: It was not possible to review the Python program.
Minor corrections
Line 3 : are have also been improved =>?? have also been improved
Line 79 : The proof of Theorem 3 about our Cuproof will be given in Appendix A. => Line 205 : ?? Appendix F.
Line 194 : In this paper, we construct an kind of range proof scheme => In this paper, we construct a kind of range proof scheme
Appendix : In the Arregating Logarithmic Proofs => In the Aggregating Logarithmic Proofs
Author Response
Dear Anonymous Reviewer,
Thank you vey much for your valuble comments on our manuscript, and we have modified it according to your valuable suggestions. If you would like to review the Python program for our implementation on Cuproof and we will send you our Python program.
Best Regards,
Lin You

Reviewer 2 Report
The manuscript, titled as "Cuproof: Range Proof with Constant Size", is an interesting study to explore the research possibility of the zero knowledge proof which can be beneficial to a number of cryptographic areas, including blockchain. I believe this study is a good addition to the journal after the revision as follows:
- The relationship between this manuscript and the journal/special issue should be explicitly stated. How can your proposed scheme contributed to the area of "Information Theoretical Security and Privacy". Additional references should be cited, for example:
- How does overconfidence affect information security investment and information security performance?. Enterprise Information Systems, 15(4), 474-491. (2021).
- Exploring the intellectual cores of the blockchain–Internet of Things (BIoT). Journal of Enterprise Information Management. (2021).
- A survey on zero-knowledge proof in blockchain. IEEE Network, 35(4), 198-205. (2021).
- It is clear that authors have proposed a novel scheme based on the zero knowledge proof, but practical implications are limited. Authors should elaborate more about assumptions, potential application areas, preceived benefits etc.
- Limitation and future work from this study should be further elaborated.
Author Response
Dear Anonymous Reviewer,
Thank you very much for your valubale comments, we have revised our manuscript according to your valuable suggestions, and the revised parties have been marked in blue. We have modified our paper with reference to the papers you recommended which have been added in our reference list. We have also given a table (Table 2) for the comparison performances of our Cuproof compared with other four zero-knowledge proofs.
Best Regards,
Lin You